# Buying Game Derivative Products Is Different from In-Game Purchases: A Mixed-Method Approach

**DOI:** 10.3390/bs14080652

**Published:** 2024-07-28

**Authors:** Xinyi An, Yuhan Peng, Zexi Dai, Yunheng Wang, Zizhen Zhou, Xianglong Zeng

**Affiliations:** Beijing Key Laboratory of Applied Experimental Psychology, Faculty of Psychology, Beijing Normal University, Beijing 100875, China; 202111998221@mail.bnu.edu.cn (X.A.); 202111998305@mail.bnu.edu.cn (Y.P.); 202211061077@mail.bnu.edu.cn (Z.D.); wangyunheng2020@outlook.com (Y.W.); zhouzizhen_99@foxmail.com (Z.Z.)

**Keywords:** derivative products, in-game purchases, character attachment, game loyalty, impulsive buying

## Abstract

The consumption of games has received increasing attention due to their high profits and addiction issues. However, previous studies have focused mainly on players’ in-game purchases, neglecting the purchase of game derivative products. This article provides the first exploration of the differences and similarities between in-game purchases and derivative product purchases with a mixed-method approach. A quantitative survey collected data from 9864 game players, and the results suggested that there were differences between in-game purchases and derivative product consumption in terms of consumption amount and number of participants, and that derivative product purchases had a stronger relationship with character attachment and game loyalty. Subsequent interviews were conducted with 22 players. The findings supported the quantitative results and revealed that players exhibited a distinct understanding of each type of purchase in terms of ownership. Overall, these findings contribute to the understanding of game derivative product purchases and explore the factors influencing in-game purchases and derivative product purchases. We strongly argue that the pattern of spending on game derivative products is very different from that of spending on in-game purchases and is, thus, worthy of dedicated research.

## 1. Introduction

In recent years, the gaming industry has generated enormous revenue, with the online game market experiencing rapid growth. It has been estimated that the projected revenue in the online game markets worldwide is expected to reach USD 27.97 billion in 2024 [1] while the most vital source of revenue for game developers is in-game purchases [2]. Online games refer to “games that are played over the Internet using PCs and game consoles” [3], and are one sort and a category of entertainment which is oriented based on information technology adoption [4]. It can be a sort of creative activity in which many games warrant a special type of concentration as well as a type of interaction in which the gamer helps to create various narratives by directly affecting the plot [5]. Players can obtain virtual items by in-game purchases, including weapons, characters, and clothing, to increase the enjoyment of the game [6,7,8]; these purchased items are essential materials during game playing, especially in role-playing games where players assume the roles of characters in a fictional setting and interact with them [9].

The issue of in-game purchases has received extensive attention from researchers. Studies have explored some factors that influence players’ in-game purchases [10,11,12]. For instance, players’ perceived enjoyment [13,14,15], the parasocial relationships between players and characters [16], and others’ attitudes toward buying virtual goods [14] influence users’ purchase intentions. Some players also pay to improve their characters’ competency [17]. Other behaviors, such as online game loyalty and problematic gaming, were found to be related to in-game purchases [2].

While previous studies have focused on understanding in-game purchases, players are also enthusiastic about derivative products outside the game. Derivative products are those that are based on or derived from an existing work, but independent of the original form [18]. Derivative products encompass all kinds of creative work. They include dolls and toys derived from animations, mobile games and movies [19], and prints, key chains, postcards, and clothing derived from art works [20]. In other words, a derivative work is an extension of an existing product [18]. In service-intensive industries such as animation, manga, games, and tourism, derivative products can be developed using existing services, brands, or cultural resources [21]. These game derivative products such as pillows, cosplay costumes, figurines, concept art books, badges, and so on [22] have become increasingly popular in recent years and have generated significant revenue for related industries [23]. For instance, the official setting collection of Arknights generated sales revenue exceeding 25.9 million RMB for its developer, HYPERGRYPH, within just one hour of its release [24]. Another developer, miHoYo, has even launched derivative product manufacturing services to integrate the derivative product industry chain [25]. Sheu et al. [19] also mentioned that derivative products can extend the related industry to earn more business opportunities.

The consumer market for game derivative products is enormous, yet limited research has focused on the consumer behavior of derivative products outside games. That is, the desire to purchase game derivative products is related to the game experience and the quality of service [26]; it could be influenced by the degree of pleasure in communicating with peers [27] and information and identification from peers [19]. Additionally, Niu et al. [28] found that the main factor influencing Japanese adolescents’ purchasing intentions in terms of animation, comics, and game (ACG) derivative products is their strong interest in and willingness to participate in ACG. However, although some researchers have noted this phenomenon, the literature on game-related purchases has neglected to mention how derivative product purchase behavior differs from in-game purchase behavior—a question that remains unanswered. Liu and Lai [22] argued that derivative products were in great demand among players who hoped to connect themselves with their favorite characters in reality, which is distinct from in-game purchases. However, to our knowledge, no research has explored the differences and similarities between purchasing game-derived products and in-game purchases, such as the amount of spending and the psychological factors that relate to preference for the types of consumption.

Since the study of game derivative products has received relatively little attention from researchers, the present work can be considered as an initial exploration. By comparing in-game purchases with the consumption of derivative products, the current research tries to gain a better understanding of consumer purchasing decisions and behavior patterns across contexts. A mixed-method design was applied in this research. With a cross-sectional survey collecting quantitative data, and qualitative interviews explaining and refining the findings presented in the survey, the current research facilitated a better understanding of the differences and similarities between the two types of consumption, and how players perceive in-game purchases and derivative product purchases.

### 1.1. Key Literature Review and Hypotheses

The similarities and differences between in-game purchases and the purchases of derivative products have important research significance.

First of all, research on the differences and similarities between these two consumption patterns can help interpret the game derivative products with the existing research results of in-game purchases and better analyze the consumers’ purchase of game derivative products from the perspective of consumers’ psychological needs. At present, the academic research on derivative products is relatively limited. In contrast, the research on in-game purchases has attracted extensive attention from researchers [5,10,13,14,29,30,31,32]. Therefore, to understand why consumers buy derivative products, a better starting point is to compare them with in-game purchases. The market for derivative products is huge, but if playing a game itself is sufficiently enjoyable, why would anyone choose to spend money on derivative products outside of the game rather than spending it all on in-game purchases? Are there some of the same or different factors that more relate to in-game purchases or derivative products? Understanding the similarities and differences between in-game purchases and derivative product purchases can help game developers better understand players’ spending behavior and preferences so that they can better design games and related products to improve user satisfaction both inside and outside the game.

Secondly, understanding the similarities and differences between in-game purchases and derivative product purchases can also provide valuable market research insights into game mechanisms and marketing strategies. For game developers, the marketing of in-game content can bring huge profits and have an impact on game design strategy [33]. Game developers will try to make games in a way that attracts users to buy game content as often as possible [34], with the aim of selling in-game content rather than just trying to make the best game in an artistic sense. However, many in-game purchase strategies induce excessive consumption and have a negative impact on players which has been criticized and restricted by local government departments [35]. If the game industry wants to develop better, it should better understand the needs of the consumers and create profit points through other ways to promote the healthy development of the industry. Actually, as we look beyond just in-game purchases, in recent years, the market of game derivative products has gradually expanded, and the scale of China’s online game derivatives market reached 3.5 billion RMB in 2022 [36], which also indicates that the game derivative market has the potential to become a new profit point. By understanding consumers’ needs and preferences in this emerging market, developers can create innovative and appealing game derivative products that resonate with players and contribute to the overall growth and sustainability of the gaming industry.

There are some drawbacks to mixing in-game purchases with the purchases of game derivative products. First of all, it will cause consumers to be unable to accurately judge the difference and value of the two consumption modes when making purchase decisions, so they may blindly follow the trend and waste money. Secondly, game developers cannot clearly understand the real needs and preferences of players, and cannot effectively meet the needs of players, which affects the quality and market competitiveness of games and products. Therefore, it is important for the game industry and consumers to distinguish between in-game consumption and the purchase of game derivative products, which helps promote the development of the game industry and optimize the consumer experience.

To identify the factors that may differ between derivative product purchases and in-game purchases, user-related variables, game-related variables, and consumption amount variables were analyzed based on brand loyalty theory as the research framework [29]. Brand loyalty theory illustrates the factors that relate to brand loyalty. Specifically, customer and product attributes such as emotion and attitude can affect brand loyalty [29]. The theory has been employed to better understand customer’s purchasing behavior, and some researchers have also used the theory to study online game consumption [30]. Our current research, as an exploratory effort, is the first to apply this theory to compare the buying behaviors of game derivative products and in-game items.

User-related variables in the current research include game character attachment, problematic gaming, and impulsive purchases; these factors are mainly related to the emotions of players. Players use characters to enter the virtual world in role-playing games, which serves as a means to explore alternative realities [31], and leads to a strong emotional connection with characters [32], referred to as “character attachment” [37]. And the relationship between players and their avatars can influence overall game satisfaction [38]. Problematic gaming behavior is also related to emotions. In the context of online games, many studies have reported that games are used more by players for emotional regulation purposes [39,40]. Estevez et al. (2017) [41] showed that the lack of emotional control is indicative of problematic gaming in the sample. In addition, studies have shown that emotional stimuli are also a key factor in impulsive purchases [42]. At the same time, research based on other theories such as customer value theory also mentions that emotional value can positively affect purchase intentions toward digital items [43]. So, it makes sense to pay attention to emotional factors in in-game purchases and derivative product purchases.

To analyze game-related variables with respect to game items, the RFM model (recency, frequency, and monetary) was taken into account. The RFM model is frequently used to analyze customers’ use patterns and loyalty [44]. R (recency) denotes the latest purchase date; the closer a customer makes a purchase, the more likely he or she is to make another purchase. F (frequency) indicates the number of purchases during a specific period, and the higher the frequency of purchase, the stronger the loyalty. Finally, M (monetary) represents the purchased amount within a specific period of time, and the higher the amount of purchase expenses, the stronger the loyalty [30]. We chose the game frequency for our analysis because the frequency-related variables, such as the purchase period or cycle, have stronger effects on the repurchase of online game items [30]. And an event frequency based on data representation can be used to predict multiple player behaviors [45]. This behavior is defined in the literature as “habit” [46], which also has a significant effect on loyalty. As for the R factor and the M factor, due to the diverse range of packages available for in-game purchases offered by game developers, some customers may not make frequent purchases, but the value of each purchase could be substantial. Solely considering the timing of the last purchase may underestimate the significance of these customers. Therefore, the recency of purchases is not taken into account in this study. The purchase expenses associated with the two types of consumption are considered as our dependent variables.

By comparing the spending amounts and number of participants in in-game purchases from derivative product purchases, we aim to delineate the fundamental distinctions between these two types of game-related purchase behaviors first (H1). We then theoretically explain the correlation of the two types of game-related purchases from the perspective of emotion and behaviors (H2a–H2d, H3a). Finally, we demonstrate that the correlations between players’ emotional and behavioral factors and the two types of game-related purchase behaviors are linked to game loyalty (H3b). Figure 1 depicts the interrelationships among these major constructs.

#### 1.1.1. Differences in the Number of Consumers and Spending Amount

Although both are related to games, there are primary differences between derivative products and in-game purchases, such as the amount of money spent and the number of players who prefer a particular purchase. For instance, Liu and Lai [22] found that while some players spend time and money only inside the game, some players invest time and money both inside and outside the game; they did not delve further into the reasons and specific differences in amount. Since in-game purchases allow players to purchase features and virtual items that have functional or cosmetic value [34], giving players a performance advantage during gameplay, players may prefer in-game purchases to enhance their experience and satisfy their autonomy and competence needs [47]. This preference may be attributed to the fact that the majority of the individuals involved in online gaming are focused on the gaming experience [48,49], while such benefits for game experience are not applicable if players spend money on derivative products. Therefore, players should prioritize in-game purchases rather than spend on derivative products to maximize their gaming experience. Furthermore, the channels of in-game purchases are often designed in the game interface, which makes them easier for players to access than derivative products that are sold on shopping websites or in offline stores, which may make players more enthusiastic about in-game purchases. So, the following hypothesis is proposed:

**Hypothesis 1 (H1).** 
*More people make in-game purchases than consume derivative products, and players spend significantly more on in-game purchases than on derivative products.*


#### 1.1.2. Character Attachment

Players use characters to enter the virtual world in most online games, and this leads to a strong emotional connection with characters [32], referred to as “character attachment” [37]. Studies have shown that emotional attachment to virtual characters can positively affect game-related purchase intentions. For in-game purchases, as players develop emotional connections with their characters and become more attached to them, they are more likely to buy virtual items to enhance their character’s performance and abilities [50,51]. An empirical study confirmed that the most common motivation for in-game purchases is to acquire characters or rare items [52]. In addition, players are willing to pay for their favorite characters because they view investing money as a way to support their favorite characters and obtain emotional satisfaction [22]. Researchers have found that derivative products are in great demand among fans, which reflects the emotional demands of players to connect with their favorite characters in reality [22]. Therefore, character attachment level should have positive correlations with both in-game purchases and derivative product purchases.

Although the consumption of game derivative products may not enhance the gaming experience, we believe that there may be a stronger correlation between character attachment and derivative products than between character attachment and in-game purchases because derivative products enable players to bring the character image into the real world, offering more emotional companionship and connection. Thus, the emotional attachment to game characters may have a stronger correlation with the players buying derivative products outside the game. The following hypothesis is proposed:

**Hypothesis 2a (H2a).** 
*Players’ degree of character attachment is positively correlated with both in-game purchases and derivative purchases, and the correlation with derivative product purchases is greater than that with in-game purchases.*


#### 1.1.3. Weekly Game Frequency

It is common to reward players for logging in daily with rare items or free in-app purchasable items in free-to-play games. Research has shown that game frequency is positively correlated with in-game purchase intention, and the more frequently players play, the more likely they are to purchase virtual items [53]. Therefore, the same hypothesis is made in this study: weekly game frequency is positively correlated with in-game purchases.

However, the purchase intention for virtual items may not translate to derivative products. Frequent game logins show high engagement in the virtual world, but buying derivative products does not enhance the virtual experience, potentially reducing player interest in those products. Additionally, many virtual items in games are consumables that disappear or become ineffective after use and need to be purchased continuously; therefore, the frequency of game playing may be more related to in-game purchases, while many derivative products often do not require repurchasing. Therefore, the following assumption is made:

**Hypothesis 2b (H2b).** 
*Players’ weekly game frequency is positively correlated with their in-game purchases, and the degree of correlation with in-game purchases is greater than that with derivative products.*


#### 1.1.4. Problematic Gaming

Problematic gaming is a persistent, pathological pattern of online gaming behavior that may result in impaired functioning and clinical distress [54,55,56]. Previous studies have shown that problematic gaming is associated with more in-game purchases [57], and that the level of problematic gaming is a determinant of in-game purchases [58]. Players with problematic gaming may have low self-esteem and redefine their self-worth according to their game-playing abilities and investment in games [59]. As spending outside the game cannot improve game performance and experience, problematic gaming behaviors may not be strongly correlated with purchasing derivative products. Therefore, the following hypothesis is proposed:

**Hypothesis 2c (H2c).** 
*Problematic gaming is positively correlated with in-game purchases, and the degree of positive correlation is greater than that with purchasing derivative products.*


#### 1.1.5. Impulsive Buying Tendency

Impulsive buying has been described as “a consumer’s tendency to buy spontaneously, unreflectively, immediately, and kinetically” [60] (p. 306). Studies have shown that game players’ purchase intentions for probabilistic items are related to both rational and impulsive factors [61]. To stimulate players’ paying urges, game developers often adopt various marketing strategies, such as introducing time-limited discounts or rewards. Moreover, in-game purchases often occur within the game, and as players are immersed in the emotional experience of the game, it is reasonable to predict that they are more likely to make irrational purchases. Therefore, we hypothesize that the player’s impulsive buying tendency has a strong positive correlation with in-game purchases.

However, as derivative products are often bought outside of game time, consumers may tend to be more rational regarding these purchases. A series of processes and factors, such as the purchase channels of derivative products outside the game, the efforts made to select goods, and the waiting time for the mailing process, may also allow consumers to return to rational and less impulsive mindsets. Therefore, although players’ impulsive buying tendencies may also have a positive correlation with derivative product purchases, it should be weaker than the correlation with in-game purchases. Therefore, the following hypothesis is proposed:

**Hypothesis 2d (H2d).** 
*Players’ tendencies to engage in impulsive buying are positively correlated with in-game purchases, and the degree of this positive correlation is greater than that for purchasing derivative products.*


#### 1.1.6. Game Loyalty

Customer loyalty is defined in marketing as the customer’s repeated use of a specific company, store, or product [62], which is considered to be the highest level of relationship among all the consumer–brand relationships [2]. Some studies have regarded loyalty as a function of regulating purchase behavior and other important results [63], which is critical for long-run profitability [64]. Game loyalty has always been a research hotspot in this field with the gradual saturation of the market and the increase in competitors [65]. Studies have shown that the willingness to continue playing a game is positively correlated with in-game purchase intentions [14,16,66], and players’ game loyalty positively predicts in-game purchase intentions [10,34,67,68,69,70,71,72]. Considering that there is no strong evidence for how game loyalty relates to derivative product purchases and whether this factor is more influential for derivative products purchases compared to in-game purchases, we presume that players’ game loyalty is positively correlated with both.

**Hypothesis 3a (H3a).** 
*Players’ game loyalty is positively correlated with both in-game purchases and purchasing derivative products.*


Game loyalty is also influenced by the variables mentioned earlier in this study. Specifically, players’ attachment to a specific character could lead to greater loyalty to that game [73]. Regarding game frequency, researchers have argued that prior usage frequency helps individuals form habits, which in turn positively impact game loyalty [74]. Players who play more frequently each week tend to develop habitual gaming behavior and show greater loyalty to the game. For problematic gaming, studies have found that problematic gaming can lead to loyalty toward the game [75], and Balakrishnan and Griffiths [2] found that game loyalty mediates the relationship between problematic gaming and purchase intention. Given that the players’ game loyalty positively predicts in-game purchase intentions [10,34,67,68,69,70,71,72], it is expected that game loyalty will mediate the relationship between behavioral and emotional factors and consumption. By examining the variance in the mediating effect of game loyalty between the two types of consumption and emotional and behavioral factors, we can gain insight into the difference between in-game purchases and game derivative product purchases, uncover the distinct characteristics and patterns associated with each type of game-related consumption, and elucidate the decision-making process and consumer propensity when faced with the choice between in-game purchases and game derivative products.

Therefore, we hypothesize that game loyalty might further mediate the influence of the aforementioned factors on game-related purchase behaviors. The proposed model is depicted in Figure 1. However, the theoretical foundation for the relationship between impulsive consumption and game loyalty is lacking, so it is not included in the model. Age, gender, and income were controlled for socioeconomic factors [10].

**Hypothesis 3b (H3b).** 
*The game loyalty of players mediates the effects of weekly game frequency, problematic gaming, and character attachment on in-game purchases and derivative product purchases, and there are significant differences in the effects of these three factors on in-game purchases and derivative product purchases.*


## 2. Materials and Methods

### 2.1. Study Design

A multi-method research design was applied in this study [76,77]. First, an online cross-sectional survey as Study 1 collected data on purchase behavior and other variables, such as game frequency, character attachment, problematic gaming, and impulsive consumption intention, aiming to explore the primary differences and similarities between these variables. The dependent variables are the players’ consumption amount of the two types of purchases. Then, since the quantitative results revealed only a statistical correlation, qualitative interviews as Study 2 were utilized to explain and refine the findings presented in the survey and further facilitated a better understanding of how players perceive in-game purchases and derivative product consumption.

### 2.2. Participants

In the quantitative component of the research, we collected data from Genshin Impact players (a popular role-playing game (RPG)) through an online social network. Informed consent was obtained from each participant before data collection. A total of 11,763 responses were collected. After filtering out incomplete responses and responses that did not satisfy the screening criteria (e.g., failed attention checks), the final sample consisted of 9864 responses. The demographic characteristics of the participants are detailed in Appendix A. The gender distribution is basically consistent with the findings of previous studies on RPGs and other types of online games [7,78], indicating that it is acceptable.

Following an explanatory sequential mixed research approach [79], we selected 22 participants from those who completed the questionnaire in the quantitative component above for online semistructured interviews, comprising 16 males (72.7%) and 6 females (27.3%). The selection criteria included consenting to follow-up, having purchased any in-game or derivative products, and possessing a significant period of engagement and personal understanding of the game. The interviews commenced formally after reading an informed consent statement and obtaining verbal consent from the interviewees. The participants’ demographic profiles are shown in Appendix A. All the objectives of the research were explained to the participants in detail. The study was approved by the Institutional Review Board of the Faculty of Psychology, Beijing Normal University (n. 202305160087).

### 2.3. Instruments

#### 2.3.1. Survey

Character attachment was measured with 4 items sourced from Ko [50]. A 7-point Likert scale was used with scores ranging from strongly disagree to strongly agree. An item example is “I feel very affectionate toward my favorite game character”.

Problematic gaming was assessed using eight items adapted from Pontes et al. [80]. An item example is “I often neglect many things around me because I’m so focused on playing games”. The items were measured using 5-point Likert-type scales ranging from “Strongly Agree” to “Strongly Disagree”.

Impulsive buying tendency was assessed using 26 items adapted from Jing et al. [81]. The items were measured using 5-point Likert-type scales ranging from “Strongly Agree” to “Strongly Disagree”. An item example is “I’ve noticed that I can usually resist the urge to make purchases”.

Game loyalty was measured with 2 items adapted from Choi [62], with the name of the game replacing the original keyword. The items were measured using 5-point Likert-type scales ranging from “Strongly Agree” to “Strongly Disagree”. An item example is “Genshin Impact was overall satisfactory enough to replay later”.

In-game purchases and derivative product purchases were measured by asking the participants to estimate the total spending on in-game purchases and derivative products until the present.

Game frequency was obtained from a single item in the questionnaire (“How often do you play Genshin Impact in a week recently?”). The above items are shown in the Appendix A.

Table 1 shows the relevant indexes related to the quality of the data. Before proceeding with the structural equation model analysis, an assessment of potential multicollinearity among the independent variables was conducted. The composite reliability (CR) values are greater than the threshold value of 0.70 [82] (pp. 295–358). The internal consistency reliabilities are examined using Cronbach’s alphas and range from 0.770 to 0.912, which are all higher than the recommended value of 0.5 [83].The variance inflation factor (VIF) values of character attachment, problematic gaming, impulsive buying tendency, and game loyalty range from 1.088 to 1.139, which are lower than the requirement of a VIF less than 5 [84]; therefore, multicollinearity is absent.

#### 2.3.2. Interview

The interview outline for this study was developed based on the data analysis results from the survey and consisted of four sections: game data, game spending, game characters, and game evaluation. Tencent Meeting was used for online interviews, and local audio recording was enabled. Most interviews lasted between 40 and 80 min.

### 2.4. Analysis

#### 2.4.1. Survey

SPSS 27.0 was used to calculate correlations. AMOS 28.0 was employed for structural equation modeling (SEM). A logarithmic transformation was applied to address the issue of large variance and extreme values in spending data after adding ten to the raw expenditure values, which ensured that the cases with zero spending could be logged.

#### 2.4.2. Interview

The audio recordings were transcribed into written transcripts using NVivo11 for coding and analysis applying the Theme Analysis method and combining the research design logic of interpretive sequential design.

## 3. Results

### 3.1. Quantitative Results

#### 3.1.1. Differences in Number of People and Spending Amounts

Table 2 presents the descriptive statistics of the purchase behaviors. First, the number of people making in-game purchases (n = 8844) was approximately 2.3 times the number of people who engaged in derivative product purchases (n = 3842). This indicates that in-game purchases are more prevalent. It is also notable that there are 240 players (2.43% of the total sample) spent money only on derivative products without any in-game purchases among the 3842 consumers who have bought derivative products. The data results also showed the difference in the spending amounts. The paired t-test indicated that the amount of money spent on in-game purchases is significantly greater than that spent on derivative products for the full sample (t = 26.541, *p* < 0.001) and for the sample with both in-game purchases and derivative product purchases (t = 17.135, *p* < 0.001). H1 is well supported.

#### 3.1.2. Correlations between Purchase Behavior and Other Correlated Variables

Table 3 presents the correlations between the variables (i.e., game frequency, character attachment, problematic gaming, and game loyalty) and two types of purchases (i.e., in-game spending and derivative product spending; with translation of log + 10). Character attachment positively correlates with both derivative products and in-game purchases, with a stronger correlation observed for derivative products purchases, supporting H2a. Both weekly game frequency and impulsive buying were more strongly correlated with derivative product purchases than with in-game purchases, supporting H2b and H2d. Game loyalty was significantly correlated with the two types of purchases, supporting H3a. Unexpectedly, problematic gaming did not show significant correlations with the two types of purchases; H2c is not supported.

#### 3.1.3. Tests of the Mediating Effects

This study predicts an indirect effect of the above factors on the spending amounts for the two types of purchases, as serially mediated by game loyalty (see Table 4). The standardized path coefficients for the hypothesized model are presented in Figure 2. The 95% CIs and *p* values are shown in Appendix A. For H3b, the results indicated that game loyalty mediated the relationship between weekly game frequency and in-game purchases (β = −0.003, *p* = 0.031, 95% CI = [−0.005, −0.000]) and between game frequency and derivative product purchases (β = 0.005, *p* < 0.001, 95% CI = [0.003, 0.007]). It also mediated the relationship between character attachment and the two types of purchases (in-game purchases: β = −0.006, *p* = 0.031, 95% CI = [−0.011, −0.000]; derivative product purchases: β = 0.010, *p* < 0.001, 95% CI = [0.005, 0.014]). Importantly, we found differences in the mediating effects of game loyalty on in-game purchases and derivative product purchases for weekly game frequency (β = −0.007, *p* < 0.001, 95% CI = [−0.010, −0.005]) and for character attachment (β = −0.016, *p* < 0.001, 95% CI = [−0.022, −0.010]). That is, players who would like to continue playing due to attachment to characters and with higher weekly game frequency were more likely to pay for derivative products, providing support for the potential differences in the mechanisms of the two purchases.

Using bootstrap resampling, the model fit was evaluated by reference to the comparative fit index (CFI), the goodness-of-fit index (GFI), the adjusted goodness-of-fit index (AGFI), and the root mean square error of approximation (RMSEA). The results of the current study indicate that the model fit is suboptimal, with CFI = 0.619, GFI = 0.960, AGFI = 0.790, and RMSEA = 0.179. While the model’s predictive capacity is limited, the observed trends may be indicative of underlying mechanisms not fully accounted for in the current model. Despite the model’s limitations, the exploratory analysis revealed some intriguing results as mentioned above.

### 3.2. Qualitative Results

The purpose of Study 2 was to conduct a more in-depth analysis of the results obtained in Study 1 through interviews. We wanted to understand the propensity of players to spend money on their favorite characters. Specifically, we wanted to know the following: For in-game purchases and derivative products, what type of spending are players more likely to make? How do they perceive the differences between the two types of purchases? Since Study 1 only shows statistical correlation and cross-sectional relationship between variables, Study 2 will further explore the players’ subjective perception through interviews to verify part of the results of Study 1 by directly asking the participants to share their opinions freely, and explore the players’ subjective views on the similarities and differences between the two types of consumption, as well as other factors that influence the participants’ purchasing tendencies. In addition, in Study 1, we did not incorporate impulsive purchases into the model, but impulsive in-game purchases are still a common phenomenon, particularly in the context of in-app purchases [85,86,87]. Therefore, in Study 2, we expanded the investigation to explore the conditions under which players are likely to make impulsive purchases, as well as whether consumers can recognize when they are recharging impulsively in the game.

#### 3.2.1. Theme 1: The Difference in Spending Amounts between In-Game Purchases and Derivative Product Purchases

Through interviews, we identified three types of players. The majority of players made only in-game purchases, while a second group not only made in-game purchases but also invested in game-related derivatives. A very small number of players solely bought game derivatives. We also found that while most participants (20 out of 22 players) spent more on in-game purchases than on derivative products, 2 participants primarily engaged in derivative purchases, and they explained the reasons for this. For example, P21 spent more than 3000 yuan in the game but 5000 yuan outside the game.


*“Spent over 3000 yuan (in the game), for my favorite characters… Ganyu’s garage kit has arrived, and some of her acrylic stands and so on. I also participated in the collaborations, and I also had a poster and a headset to collaborate with Xiaomi. Spent about 5000 yuan”.*
[P21]

#### 3.2.2. Theme 2: Character Attachment and Game Purchase Intention

Among the reasons for game-related purchasing, most participants said that in-game purchases and derivative product purchases are both due to character attachment. This was consistent with the quantitative results in Study 1. Emotional commitment to and affection for characters increase players’ investment in characters, including time, money, and emotion.


*“Like buying Klee’s cup. In the game, I bought outfits for Keqing and summoned weapons for Eula”.*
[P17]


*“Like, if there’s a derivative of the game that’s related to them, I’ll give it priority”.*
[P18]


*“Yes, pull for the characters and summon the weapons for them”.*
[P11]


*“Resources are tilted toward favorite characters”.*
[P14]


*“Will invest more time, such as the two months already spent on Alhaitham”.*
[P1]


*“For my favorite Raiden Shogun, I saved about 180 intertwined fates (a kind of token in the game); other characters, I could not save that much”.*
[P14]

Besides, some participants consider that investing time and money in a character will, in turn, increase their affection towards the characters. However, some other participants believe that the increase in the degree of attachment is limited, or even non-existent.


*“Yes, the more time and money invested, the more likely one may come to attach to them (the characters)”.*
[P1]


*“I might consider it based on the situation to see if it can improve my gaming experience. If it does, I would prefer it”.*
[P7]


*“I might use this character more often in the game, which could potentially deepen my fondness for it, but maybe not to such a great extent”.*
[P22]

#### 3.2.3. Theme 3: The Players’ Perceived Differences between In-Game Purchases and Derivative Product Purchases

The results of the survey revealed that there are significant differences between in-game purchases and derivative product purchases. Further interviews confirmed that players had different understandings of in-game purchases and derivative product purchases, and they also had different experiences.

Some participants were more concerned about in-game improvement and believed that derivative products were unnecessary.


*“If I buy an acrylic stand, or buy a pendant, it might have no use to me, but if I put the same amount of money into the game, it might have an intensity boost”.*
[P22]

However, in other participants’ responses, in-game purchases were often viewed as virtual, disembodied items, while derivative product purchases were more real, physical, and deterministic.


*“When you spend money in a game, the more money you charge, the less sense you feel. …. But, out of the game, although the delivery is slow, if you bid on a game derivative, I think it is better to spend out of the game than in the game”.*
[P1]


*“It feels more deterministic to buy game derivatives than to spend money in games. Buying derivative products is more similar to that if I love a character, I want to touch it and have company”.*
[P4]

Additionally, some participants mentioned that the satisfaction and sense of ownership brought about by the two types of purchases could also be different.


*“Because some people overspend in the game, there may be some negative emotions. Buying game derivative products probably not”.*
[P4]


*“In the game, because it is a combination of money and time, it has a sense of accomplishment. But, if it is a derivative of the game, the sense of gain comes in the first few days, and then it is gone”.*
[P21]

#### 3.2.4. Theme 4: Impulsive Buying

While the survey revealed a correlation between impulsive shopping traits and purchases, the interviews indicated that the participants could feel the impulsiveness or irrationality they experienced when purchasing. The reason for some participants’ impulsive consumption was that they did not have the character or weapon they wanted for a long time, which was influenced by their affection for the character mentioned above. The participants’ strong emotions may prompt them to make impulsive purchases to obtain the character they want.


*“Absolutely yes, if I fail to get my favorite character, I’ll spend in the spur of the moment”.*
[P12]


*“When Raiden Shogun reran, honest to say, I was impulsive. But I lost 50/50 (lost a 50.000% chance to get the promotional character) in the end”.*
[P15]

While most players engaged in impulsive buying only when they missed their favorite characters, there were some exceptions among the interviewees. Some participants recalled that they had continued recharging impulsively even though they had a character or weapon they did not like.


*“I was so upset when I got Kokomi’s weapon. Then, I topped up to summon Kokomi. At that time, I did not particularly like this character, but I still summon the weapon for her”.*
[P13]

## 4. Discussion

The study findings reveal the similarities and differences between the two kinds of purchases in several aspects. The obvious commonality between the two types of consumption is that the purchase behaviors are both out of emotional affection for the game characters. Such a result emerged in the results of Study 1 and was validated in the interviews of Study 2. Nevertheless, the attachment to characters is more related to derivative product purchases. This means that when players build emotional attachments to characters in a game, they tend to purchase more derivative products than spending money in the game, which indicates that compared with in-game purchases, derivative products may play a greater role in fulfilling players’ emotional needs. It also reflected players’ emotional needs to establish connections with their favorite characters in reality [22].

For the other differences, in spending and consumer amount aspects, we found that more players engaged in and spent more on in-game purchases than on derivative products (H1). This reflects that most players spend money to improve their game experience, which is consistent with the primary profit strategy of game developers [2,53]. In the interviews, some players claimed that spending money in the game could bring immediate improvements and satisfaction, so they tended to lean toward spending money in the game; the lower engagement toward derivative products may be attributed to external factors such as slow delivery by merchants, long presale periods, the waiting time during mailing process, or players not liking the product images, among other factors.

The interviews of Study 2 also revealed that players perceived in-game purchases and out-of-game purchases to be quite different in terms of the value they bring. A few participants purchased more derivative products because they perceived them to be more tangible than in-game purchases, which they viewed as virtual and unreal. This also shows that buying game derivative products is not just an extension of in-game purchases; instead, the two types of purchases represent players’ different views on the game and different consumption needs. Buying behavior could induce feelings of wellbeing [88,89], which is also true for game-related purchases. Some players focus more on the gaming experience, while others prioritize the sense of companionship in life, which also leads to different choices to some extent. For players who prefer derivative products, they value the companionship brought by derivative products. From a marketing perspective, customers are no longer satisfied with goods or services alone [23], so the need for memorable experiences has emerged in consumer retailing and service-intensive industries [90], like the gaming industry. Therefore, this study encourages game industry practitioners and academics to capture the significance of spin-offs to players, and consider marketing strategies by attaching importance to derivative products as part of the customer experience [23].

We also noticed that the participants show a clear preference for characters they like, and this preference is reflected in their tendency to favor these characters when spending money on both in-game purchases and derivative products. That is to say, the participants’ spending is driven by their positive attitude towards the characters and the game, leading to in-game or out-of-game purchases based on this emotional connection. When discussing their favorite characters in interviews, the participants exhibit a sense of pride and dedication, conveying an attitude that anything they invest in their favorite characters is granted. Such emotions may influence their impulsive spending tendencies; when faced with their favorite characters, they are more likely to lose control, hesitate, and make decisions to spend on in-game purchases. Furthermore, from the perspective of players’ subjective feelings, the emotional experience such as the levels of satisfaction and realism brought by the two kinds of consumption is different. This may also affect their future consumption decisions. Previous research has emphasized the importance of emotional value and social value as the service’s price could influence the intention to pay for mobile Internet services [91]. These perceived values will have different effects on users’ payment behaviors in different contexts [10]. Our study showed that the effects of the values for games differ in in-game purchases and derivative product purchases, thus meriting further exploration.

Apart from the fundamental differences in the number of buyers and spending amount mentioned above, we found some factors that are more related to in-game purchases, such as impulsive spending tendencies, which is consistent with previous findings [53,61] showing that players are more rational when making derivative product purchases. Moreover, based on our qualitative study, the participants were aware of their impulsive buying tendency, and their impulsive buying was usually out of attachment to a character. Players may impulsively make in-game purchases to obtain characters or weapons they have long desired, or they may engage in retaliatory spending due to missing out on an opportunity to acquire the desired items. That is, as mentioned above, players’ affection for game characters, their impulsive buying tendency, and their different perceptions of in-game purchases and derivative products can influence where they spend more money.

Furthermore, we discovered some intriguing results beyond the initial assumptions. The quantitative results showed that 2.43% of the players only spent money on derivative products and never made in-game purchases. That is, not all players prefer in-game consumption, and in-game purchases are not a prerequisite for buying derivative products. In addition, the qualitative study showed that the participants not only spent more money on their favorite characters which is consistent with previous studies [50,51,92], but also invested more time and energy. This finding is consistent with prior research [50], demonstrating that the more players appreciate their in-game characters, the more time they invest in interacting with them and learning the story and plot behind them. Additionally, in the mediating model, game loyalty mediates the relationships among character attachment, game frequency, and game consumption, which is consistent with previous research on in-game purchases [10,14,16,34,69], partially supporting H3. It also showed that in-game purchases were similar to game derivative product purchases in the psychological mechanism mediated by game loyalty. However, game loyalty did not mediate the relationship between problematic gaming behavior and the two types of purchases, which was inconsistent with previous results [2]. This is because the present study did not establish a significant link between problematic gaming behavior and the two types of consumption in our sample (H2c). One potential reason for this disparity is that the majority of our participants were college students with limited monthly incomes. Their financial constraints may have influenced their preference for investing time rather than money in games, possibly also restricting their consumption of game-related products. Additionally, game loyalty had a negative impact on in-game purchases, which is inconsistent with the findings of previous studies (e.g., [34,70]), while indicating a positive impact on the purchase of derivative products, meaning people will spend more money on derivative products of the game they continue to play. One possibility is that such outcomes may be due to the masking effect. Since customer loyalty is defined as the customer’s repeated use of a specific company or product [62], if players are loyal to the game, they are more likely to invest time in playing the game than on in-game purchases, so the effect of character attachment on in-game purchases will be reduced.

Overall, the current research made predictions and provided strong evidence about the similarities and differences between in-game purchases and derivative product purchases. We provided empirical evidence in support of our predictions using both quantitative data and qualitative data. These findings highlighted the complex motivations behind game-related purchases, especially the consumption of game derivative products, and encouraged researchers and marketers to pay more attention to game derivative products.

## 5. Conclusions

### 5.1. Theoretical Implications

This study has made a significant theoretical contribution to the understanding of game-related purchases. First, as previous research has paid limited attention to game derivative products [19,22,26,27,28], this paper is one of the first to pay attention to them, seeking to understand the similarities and differences between people’s buying patterns for game derivative products and in-game purchases. Specifically, our findings indicated that the pattern of spending on game derivative products is very different from that of spending on in-game purchases. As suggested by their name, we assumed at first that players’ consumption of game derivative products would be an extension of in-game purchases or a secondary choice regarding game-related consumption. However, this study indicated that not all players prefer in-game consumption and that some purchased only derivative products; thus, derivative product purchases are not simple extensions, and consumers’ understanding of game derivative products and their purchase motivations greatly differ from those of in-game purchases. Through providing insight into the purchase behavior of game derivative products from the perspective of in-game purchases, this study has established exploratory evidence to understand the relationship between in-game purchases and derivative product purchase behavior.

Second, the current research provides insights into the underlying processes of how several factors, such as players’ behaviors and emotional attitudes, are associated with preferences for in-game purchases and derivative products. Previous studies have shown that emotional responses are related to consumption choices [93], and emotional responses can influence purchase behavior [94,95,96]. And young people will spend heavily on products which appeal to their emotions and create a feeling of wellbeing [97]. Our results are consistent with previous studies. The findings also demonstrate that in-game purchases and derivative products provide consumers with distinct emotional experiences, and each has its own emphasis. This research contributes to the exploration of the different and similar emotional experiences brought about by two types of consumption and a more comprehensive view for future research on game-related purchase behaviors.

Third, the results have distinguished the players with a preference for game-related consumption. Especially, the current research gives attention to these players who spend more on derivative products or only pay outside the game; they may be relatively less concerned about the virtual gaming experience and place more emphasis on emotional companionship in the real world or pay more attention to the practical functionality of derivative items beyond the digital realm. Previous studies have shown that personality traits could predict consumer behavior broadly, such as the willingness to use the service and the feeling of experience [98,99,100]. Since not all players prefer in-game consumption, perhaps there are unique personal traits among these players that differentiate them from other consumers, leading them to favor derivative products. In-depth interviews could be conducted in the future to better understand the characteristics of consumer preference groups, thus providing more detailed user profiles for the target audiences of these two types of consumption.

Finally, these findings contribute to the wider literature on in-game purchases. Some of our findings were not fully consistent with those of previous studies [2,34,70], suggesting the complexity of the relationship between in-game purchases and psychological factors. This inspires future studies to consider more factors to better capture these relationships.

### 5.2. Practical Implications

From a practical perspective, the implications of this study can be appreciated by game developers to promote or refine game product sales. First, developers can attract more players to purchase game-related content by adjusting their marketing strategies. Currently, game developers have focused on in-game purchases and created many targeting strategies, and researchers have categorized the items purchased in-game, which can help game developers provide various technical features and different types of in-game items [12]. But many have neglected the derivative product market outside the game. Thus, developers should pay attention to the consumption potential of this part of the market. Some game developers are adapting their marketing strategies for derivative products based on this trend [2,24,25,53]. By acknowledging the potential of derivative products and incorporating them into their marketing strategies, game developers can not only enhance player engagement and satisfaction with the game but also unlock new opportunities for growth and success in the industry.

Second, due to players’ surprisingly high need for building emotional connections with their favorite game characters in real life, our study indicated that derivative products should be designed and sold with a focus on the emotional needs of players. By incorporating this understanding into their marketing strategies, game developers can promote the existing derivative products and create more kinds of derivative products, and attract their target audience more effectively [12,101]. This may not only increase players’ emotional investment in the game but also open up a new revenue stream for developers.

Finally, in the rapidly growing gaming entertainment industry, developers must consider social responsibility when formulating marketing strategies and designing games and derivative products if sustainable and responsible development is to be achieved in the future. For instance, game developers should consider incorporating time limits to discourage over use to prevent players from exhibiting problematic gaming behavior.

### 5.3. Limitations and Future Directions

Through a cross-sectional study and an interview study, this research offers a snapshot of the phenomenon investigated. Future research should investigate the variables that influence the preference for in-game purchases and derivative products and the representative consumer population more comprehensively.

This study has several limitations. First, while Study 1 benefits from a broad and diverse sample, the data were self-reported using scales administered via an online survey, which might have introduced self-reporting bias. Future studies should try to obtain players’ login frequency and spending data in a more objective manner. Second, the quantitative research utilized only cross-sectional data, which cannot support conclusions about causal relationships between variables. Future studies should involve longitudinal or even experimental study designs to better explore the causal mechanisms of changes in consumption behavior. Third, to facilitate data collection, our study focused solely on Genshin Impact players, thus limiting the generalizability of the results. It may be valuable for future research to explore how players in other popular online games are influenced by similar factors in their different purchase behaviors. Additionally, as our participants in Study 2 were almost all college students with limited monthly incomes, the structure of their purchasing power and spending habits may not be representative of all the player groups. Besides, conducting interviews via online meetings may lead to less clear and direct communication, potentially causing the participants’ attention to be less focused. In future studies, it may be beneficial to try to obtain a broader range of participants in terms of age and to increase face-to-face interviews in order to make the data more comprehensive and reliable.

Since this study is exploratory, the conceptual model in Figure 1 does not stem from a complete, established theoretical framework, which is also a limitation of our current research. Firstly, since the current research does not rely on a fully developed theoretical framework of game consumption, it may have limitations in capturing the complexity of the two types of purchases under investigation. Due to the lack of clear theoretical guidance, the hypotheses of the research may not be precise enough, and the hypothetical model fit is suboptimal, leading to certain limitations in the interpretation and generalization of the research results. Secondly, the dataset, while robust in some aspects, lacks certain variables that could have improved the predictive power of the model. While the model’s fit is not ideal, the study has provided valuable insights into the relationships between the variables. In fact, game derivative products have received relatively little attention from researchers, and the value of this research lies in the exploration of game derivative purchasing behavior. Although this study has made some interesting findings on the exploratory side, these limitations need to be seriously considered in order to provide a more complete idea for future research.

## Figures and Tables

**Figure 1 behavsci-14-00652-f001:**
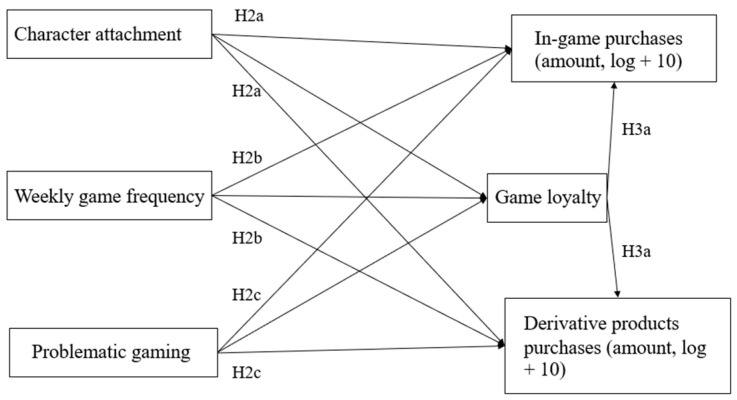
Conceptual model.

**Figure 2 behavsci-14-00652-f002:**
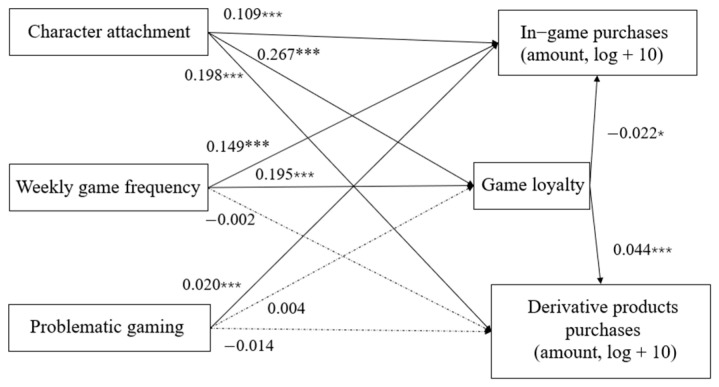
The structural equation model. * denotes significance at *p* < 0.05 level. *** denotes significance at *p* < 0.001 level.

**Table 1 behavsci-14-00652-t001:** Data quality.

Construct	CR	Alpha	VIF
Character attachment	0.914	0.912	1.139
Problematic gaming	0.776	0.770	1.137
Game loyalty	0.813	0.809	1.088
Impulse buying tendency	0.8417	0.794	1.126

**Table 2 behavsci-14-00652-t002:** Descriptive statistics of the amount of money spent within the game and outside the game.

Group	Frequency	In-Game Purchases (Amount)			Derivative Products Purchases (Amount)		
		Mean	SD	Median	Mean	SD	Median
FS	9864	3280.53	11,625.45	750	224.76	1344.37	0
IP	8844	3658.88	12,221.12	1000	244.17	1415.99	0
DP	3842	4742.36	15,484.27	1200	577.04	2106.56	200
IPO	5242	2697.25	8666.90	700	--	--	--
DPO	240	--	--	--	239.86	469.16	100

Note: FS = full sample; IP = have made in-game purchases; DP = have made derivative product purchases; IPO = in-game purchases only; DPO = derivative product purchases only.

**Table 3 behavsci-14-00652-t003:** Correlation coefficients and difference.

Construct	In-Game Purchases		Derivative Product Purchases		Difference	
	r	*p*	r	*p*	z	*p*
CA	0.055	<0.001	0.213	<0.001	−13.332	<0.001
WGF	0.183	<0.001	0.030	0.003	12.849	<0.001
PG	−0.006	0.559	0.007	0.476	−1.081	0.14
GL	0.040	<0.001	0.106	<0.001	−5.510	<0.001
IBT	0.231	<0.001	0.170	<0.001	5.219	<0.001

The correlation coefficient between in-game purchases (amount, log + 10) and derivative product purchases (amount, log + 10) is 0.287 ***. WGF = weekly game frequency; PG = problematic gaming; CA = character attachment; GL = game loyalty; IBT = impulsive buying tendency. *** denotes significance at *p* < 0.001 level.

**Table 4 behavsci-14-00652-t004:** Bias-corrected bootstrap mediation analysis with game loyalty and difference in path coefficients.

Variable	Indirect Effects								Difference	*p*
	IGP	95% CI		*p*	DPP	95% CI		*p*		
		Lower	Upper			Lower	Upper			
CA	−0.006	−0.011	−0.001	0.031	0.010	0.005	0.014	<0.001	−0.016	<0.001
WGF	−0.003	−0.005	−0.003	0.031	0.005	0.000	0.007	<0.001	−0.007	<0.001
PG	<0.001	−0.001	<0.001	0.550	<0.001	−0.001	0.001	0.668	<0.001	0.668

WGF = weekly game frequency; PG = problematic gaming; CA = character attachment; DPP = derivative product purchases (amount, log + 10); IGP = in-game purchases (amount, log + 10); the difference value is obtained by subtracting the path coefficient of OGP from the path coefficient of IGP.

## Data Availability

The datasets generated and/or analyzed during the current study are not publicly available because the dataset contains sensitive identifying information. Due to the in-place ethical restrictions, the datasets are available from the author upon reasonable request (contact Xianglong Zeng via xzeng@bnu.edu.cn).

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
