# Peer review of "Buying Game Derivative Products Is Different from In-Game Purchases: A Mixed-Method Approach"

_behavsci, 2024, doi:10.3390/bs14080652_

Round 1

Reviewer 1 Report

Comments and Suggestions for Authors

Author Response

We noticed that you mentioned the scale items in Comment 7. The scale items adopted in this study is in the supplemental material. We submitted it together with the original manuscript, and it seems that for some reasons you have not seen it. We have also added the content in the Author's Notes File.

Reviewer 2 Report

Comments and Suggestions for Authors

Dear authors,

The text is well structured. Congratulations!!

The article addresses a research gap, the methods are appropriate, and the results are particularly interesting and have the potential to provide relevant knowledge for both academics and practitioners. However, the article needs some improvements to make it stronger and more relevant to readers. Namely, in chapter 4, "discussion" you should discuss the results of your research by comparing them with other results from existing scientific literature. In this way, you will highlight your own contribution to the field of knowledge. As for references, I consider it important that you review the possibility of replacing references that are more than 10 years old with more recent ones.

Please consider these recommendations constructive based on my reading of the manuscript.

Author Response

Thank you very much for taking the time to review this manuscript. We have carefully considered the suggestion and make some changes. Please see the attachment.

Round 2

Reviewer 1 Report

Comments and Suggestions for Authors

Round 3

Reviewer 1 Report

Comments and Suggestions for Authors

The authors have improved the paper writing.
